# Changes of Physical and Mechanical Properties of Coral Reef Limestone under $CO_2$–Seawater–Rock Interaction

**Yu Zhong** [1,2]**, Qi Li** [2,3,*] **, Ren Wang** [2,3] **and Ting Yao** [2,3]

1   College of Civil Engineering and Architecture, Guangxi University, Nanning 530004, China; zhongyu1216@163.com
2   State Key Laboratory of Geomechanics and Geotechnical Engineering, Institute of Rock and Soil Mechanics, Chinese Academy of Sciences, Wuhan 430071, China; rwang@whrsm.ac.cn (R.W.); yaoting98@163.com (T.Y.)
3   University of Chinese Academy of Sciences, Beijing 100049, China
*   Correspondence: qli@whrsm.ac.cn

**Abstract:** Large amounts of anthropogenic $CO_2$ in the atmosphere are taken up when the ocean alters the seawater carbonate system, which could have a significant impact on carbonate-rich sediments. Coral reef limestone is a special biogenic carbonate, which is mainly composed of calcium carbonate. When carbonate-rich rocks are brought into contact with a $CO_2$ weak acid solution, they will be dissolved, which may affect the physical and mechanical properties of the rock. In this paper, the physical and chemical interactions between $CO_2$, seawater and the framework structure reef limestone were studied based on an experiment conducted in a hydrothermal reactor. The solution was analyzed for dissolved $Ca^{2+}$ concentration during the reaction, and the rock mass, effective volume (except for the volume of open pores), permeability, images from electron microscopy and X-ray microtomography were contrasted before and after immersion. The uniaxial compressive and tensile strength tests were conducted, respectively, to clarify the mechanical response of the rock after the reaction. The results indicate that dissolution occurred during the reaction, and the calcium ions of the solution were increased. The physical properties of the rock were changed, and the permeability significantly increased. Because the rocks were soaked for only 15 days, the total cumulative amount of calcium carbonate dissolved was less, and the mechanical properties were not affected.

**Keywords:** coral reef limestone; $CO_2$–water–rock interaction; mineral dissolution; microstructure; permeability; CT scanning

## 1. Introduction

The carbon dioxide ($CO_2$) concentration in the atmosphere has greatly increased since the beginning of the Industrial Revolution, and the emission of carbon dioxide is the main reason for global warming [1]. It has been shown that almost 30% of the $CO_2$ emitted by man is eventually absorbed by the ocean, causing ocean acidification (OA) [2], which significantly impacts the marine ecosystem [3,4]. Carbon dioxide sequestration is one of the most important means of reducing carbon dioxide emissions [5]. Marine carbon sequestration is a new kind of carbon dioxide sequestration which has much greater potential and security than terrestrial carbon sequestration [6]. However, it is a new technology that is underdeveloped, and there may be a risk of leakage due to the immature technology. The increase of dissolved carbon dioxide in the seawater leads to a decrease in pH and a perturbation of the carbonate chemistry [7]. The average pH of surface ocean has decreased by about 0.12 units since the preindustrial era and is predicted to drop by a further 0.3–0.4 units until 2100, with a pH value of 7.8 [8].

The decrease of seawater pH will cause the dissolution of carbonate minerals, and the stability of carbonate sediments may be implicated [9]. Morse et al. [10] found that the calcium carbonate-rich shallow sediments in the Behamas dissolved when the pH value

was below 8.0. Muehllehner et al. [11] conducted a broad-scale geochemical survey of across the Florida Reef Tract in 2009–2010 and found that the reefs experienced dissolution due to the acidification of water. Hall-Spencer et al. [12] studied cold vent areas off Ischia in Italy where seawater was almost acidified by $CO_2$. The researchers found that the skeletons of calcified organisms dissolved in a short time, although the pH value of seawater was about 7.3. The carbonate dissolution occurred on the surfaces, and in the nooks and crannies of the limestone foundation. Therefore, the limestone foundation became more porous and weaker [13,14]. In addition, the physical and mechanical properties of reef sediments changed with a $CO_2$ acid system, and the dissolution rates increased when OA continued [15]. Coral reef limestone is a special biogenic carbonate which is mainly composed of calcium carbonate. It is a common marine sedimentary rock and the main load-bearing medium of coral reefs [16]. Research has shown that calcite easily dissolves when brought into contact with a $CO_2$ weak acid solution [17]. Therefore, it is important to have a holistic understanding of fluid–rock interacting between $CO_2$, seawater and reef limestones.

Several studies have been devoted to the experimental investigation of the $CO_2$–water–rock interaction. Some studies have indicated that minerals of the rocks dissolve or precipitate during the process, and the physical and mechanical properties of the rock change during the reaction [18–25]. However, not all studies have yielded the same results. Angeli et al. [26] found that there was no obvious mineralization reaction when the condition was of low temperature and pressure in their tests. Similarly, $CO_2$ had no obvious effect on the tensile strength of sandstone, shale and chalk [27]. Clark et al. [28] suggested that the effects of $CO_2$-acid brines on carbonate with different types are quite different. The reef limestones are special carbonates with diverse structure types and are quite different compared to other continental and marine sedimentary rocks [29]. The effect of $CO_2$ on the physical and mechanical properties of reef limestones remains unclear.

In the current work, experiments were conducted to simulate the reaction process of $CO_2$, seawater and reef limestone to investigate the destruction of both the physical and mechanical properties of reef limestone due to the acidification of seawater. The solution was analyzed for dissolved $Ca^{2+}$ concentration during the reaction. Furthermore, the rock mass, effective volume (except for the volume of open pores) and permeability were tested before and after immersion. Uniaxial compressive and Brazilian split tensile tests were conducted to assess the mechanical properties of the samples. Multiscale imaging techniques were used to observe the changes of microstructure. The aim of this study was to deepen our understanding of the interaction among $CO_2$, seawater and coral reef limestone, especially for the evolution of the microstructure, and to determine whether the changes cause differences in the rock's physical and mechanical properties.

## 2. Materials and Methods

### 2.1. Materials

Reef limestone is usually classified as following categories: framestone, bindstone, rudstone and bioclastic limestone [30]. The coral reef limestone used in this work was a borehole core taken from a reef island in the South China Sea, and the sampling depth was 100–120 m. It is considered as framework structure according to the material composition and structure, which is composed of large coral skeletons with a higher level of density and smaller porosity compared to the other three limestone types. There were some irregular pores in these samples.

The mineralogy of the sample was measured by X-ray diffraction (XRD, D8 Advance, Bruker, Germany), and the XRD results were analyzed using Jade. 6.0. The XRD pattern of the reef limestone is shown in Figure 1. Figure 1 shows that the reef limestone was mainly composed of calcite ($CaCO_3$). The cores were cut and polished into different shapes and sizes: a cube specimen ($10 \times 10 \times 10$ mm$^3$) for scanning electronic microscopy (SEM, ZEISS Gemini SEM 300, Oberkochen, Germany); a cylinder specimen with a diameter of 25 mm and height 50 mm for the volume, permeability and uniaxial compression tests; and a disc

specimen with a diameter of 50 mm and height of 25 mm for the Brazilian test (Figure 2). The surfaces of all cube specimens were polished to make them smooth enough to obtain a better view of the changes in the surface micromorphology of the samples after corrosion. The SEM images of the surface of the cube sample before and after polished are shown in Figure 3. It can be seen that the surface of the polished sample used in this study was very flat and without irregularities particles.

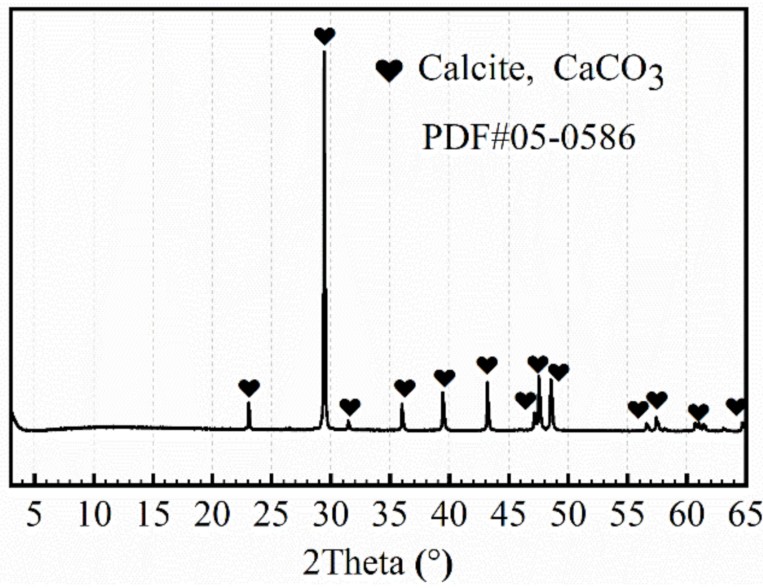

**Figure 1.** X-ray diffraction of the reef limestone.

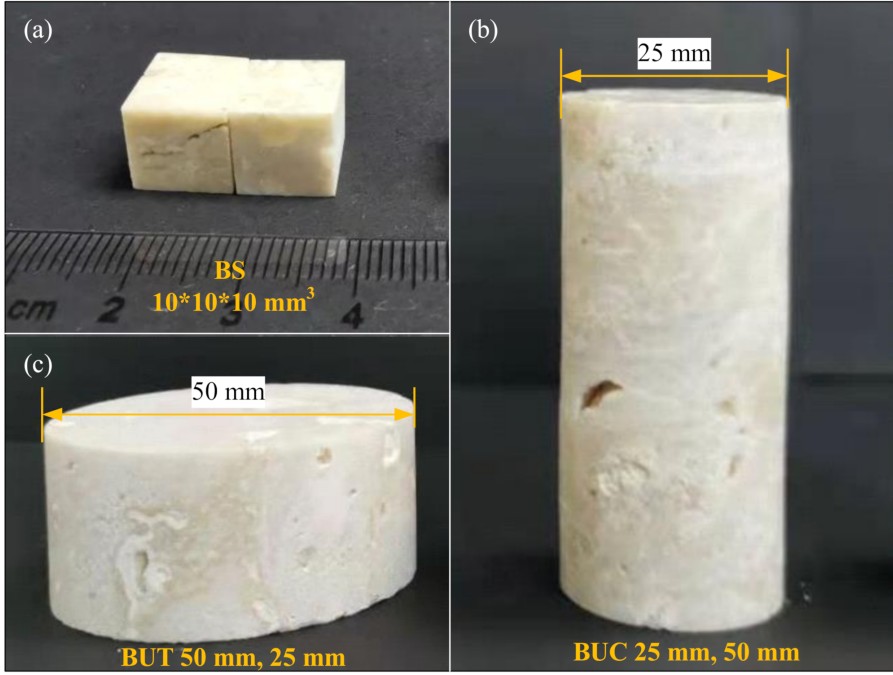

**Figure 2.** Specimens of reef limestone with different shapes and sizes: (**a**) cube specimen for the SEM test (BS), (**b**) cylinder specimen for the uniaxial compression test (BUC), (**c**) disc specimen for the Brazilian test (BUT).

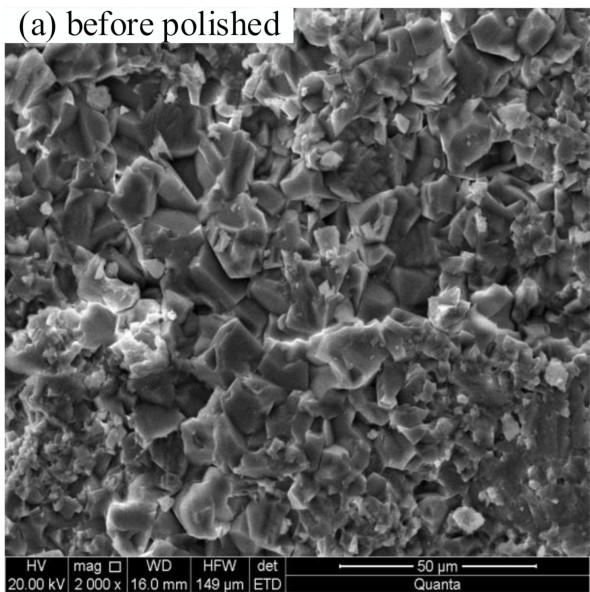
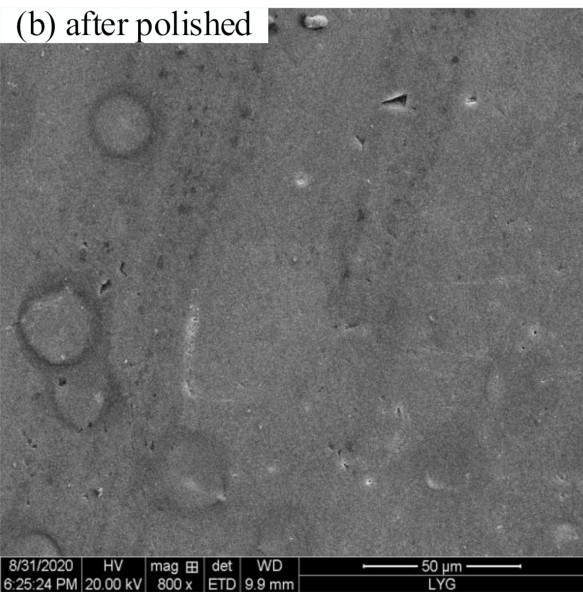

**Figure 3.** Scanning electron microscopy images of the cube samples before the reaction: (**a**) surface before polishing, (**b**) surface after polishing.

### 2.2. Apparatus and Methodology

#### 2.2.1. Water Absorption

The water absorption of the samples at ambient temperature were tested according to the DL/T5368-2007 Standards of Rock Experiment procedures of Water Conservancy and Hydropower Engineering of the Chinese Ministry of Water Resources (2007) [31]. The water absorption coefficient ($W_a$) (%) of the samples were obtained according to the following equation:

$$W_a = \frac{m_s}{m_d} \times 100\% \tag{1}$$

where $m_s$ is the wet mass of the specimen, and $m_d$ is the dry mass of the specimen. All samples were oven-dried at 105 °C for 24 h and then weighed prior to testing.

#### 2.2.2. Volume Measurement and Porosity Calculated

The length and diameter of the cylinder samples were measured by a precise vernier caliper. Then, the apparent volume ($V_t$) as calculated. The effective volume ($V_e$) of the cylinder samples, which includes the solid and closed pores [32] volumes of the sample, was measured by the AccuPyc II 1340 Gas Displacement Pycnometer (Micromeritic Instrument Corporation, Norcross, GA, USA). It is a non-destructive technique using the gas displacement method to measure the volume of material. The gas gas used in this study was nitrogen. The volume of the chamber used in this study was 100 cm³ (46.2 mm in diameter and 61.8 mm in height). The apparatus demonstrated good reproducibility, with an accuracy of ±0.03%. The effective porosity $\varphi_e$ of the cylinder samples can be calculated as follows: $\varphi_e = \frac{V_t - V_e}{V_t} \times 100\%$.

#### 2.2.3. Permeability Measurement

The transient pulse decay method appears to be one of the most applied experimental methods to determine low permeability. The principle of this method is as follows. First, the system is kept at a constant gas pressure. Then, a transient pulse is applied at the upstream end of the rock sample, which causes a pressure difference between the two ends of the rock. The attenuation relation of pressure difference with time is shown in Formula (2), and the permeability can calculated through Formula (3) [33]:

$$\frac{\Delta P(t)}{\Delta P_i} = \exp(-\alpha t) \tag{2}$$

$$k = \frac{-\alpha \cdot \mu \cdot L}{A p_m \left( \frac{1}{V_u} + \frac{1}{V_d} \right)} \tag{3}$$

where $t$ (s) is the time, $\Delta P(t)$ (atm) is the current pressure difference at time t, $\Delta P_i$ (atm) is the initial differential pressure, $\alpha$ is the exponential decay coefficient, $k$ ($\mu m^2$) is the permeability of the sample, $\mu$ (cp) is the dynamic viscosity of the gas, $L$ (cm) is the length of the sample, $A$ (cm$^2$) is the cross-sectional area of the sample, $p_m$ (atm) is the equilibrium pressure, $V_u$ (cm$^3$) is the volume of pipeline of the upstream and $V_d$ (cm$^3$) is the volume of pipeline of the downstream.

A device was developed for the measurement of permeability in this study, and its schematic diagram is shown in Figure 4. The experimental system consisted of a high-pressure nitrogen cylinder, a pore pump, a confining pump, core holder chamber, a vacuum pump, dates acquisition system, some pressure sensors, the differential pressure sensor and other essential components which were connected by high-pressure sealing pipes. The differential pressure curve, $\Delta p(t)$, was recorded by the differential pressure sensors during the test. The permeability test steps are described as follows:

(1)    The sample was wrapped in a heat shrink tube and placed vertically into the core holder. The confining pressure pump was used to deliver the ultrapure water into the pressure chamber of the core holder where a confining pressure of 1 MPa was applied.

(2)    In order to remove the air of the lines of the pore system, the vacuum pump was applied to vacuum the system for 6 h. After that, the pore pump was set at 0.2 MPa to maintain a constant sample pore pressure.

(3)    The temperature of the whole system was stabilized at the level of 30 °C using a water bath. After the whole system had stabilized (where the temperature, the pore pressure and confining pressure were stable), a pressure pulse (20 kPa) was applied. The pressure decay curve was recorded constantly by the differential pressure gauge. The permeability could be calculated through the formulas (2) and (3).

(4)    The permeability of the sample under the different confining pressures could be obtained by repeating steps (2)–(3) and changing the confining pressure in step (1). In this study, the permeabilities of the cylinder specimens were measured under the confining pressures of 1 MPa, 3 MPa, 5 MPa and 7 MPa.

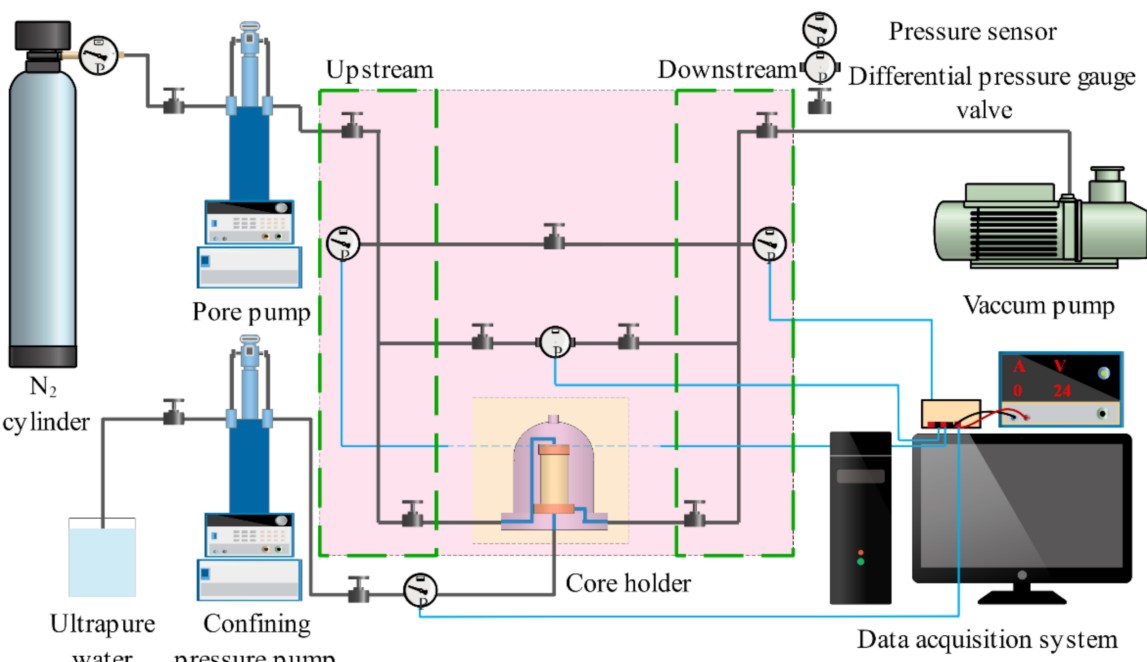

**Figure 4.** Schematic diagram of the permeability measurement system.

### 2.2.4. CO$_2$–Water–Rock Interaction

Two groups of samples were prepared for the experiments. In order to accelerate the corrosion process, rock samples of the Group CO$_2$ were interacted with the high pressure of CO$_2$ (7 MPa). Another group of rock samples was immersed in seawater in a beaker without adding extra CO$_2$, serving as a reference group. Each group contained 11 specimens, including five cube samples, three cylinder specimens for uniaxial compression test, and three Brazilian disc samples.

The CO$_2$–seawater–rock chemical interaction was simulated by exposing the rock samples in a stainless steel (316 L) hydrothermal reactor, as shown in Figure 5, equipped with a manometer and automatic temperature controller. The cylindrical reactor was fitted with a gas inlet and liquid sampling port, and the dimensions were as follows: inner diameter, 110 mm; and height, 150 mm. The CO$_2$ gas was injected by the syringe pump. In this study, the reactor was set at 30 °C, and the pressure was kept at 7 MPa. Because the salinity of seawater of the South China Sea is about 3.32–3.44% [34], the reaction solution salinity used in this study was 3.4% prepared by dissolving 34 g sea salt per liter of ultrapure water. The testing procedures for the specimens of Group CO$_2$ were as follows:

(1) The solution was prepared, and its concentration of Ca$^{2+}$ before interaction was measured by a compact calcium ion meter (HORIBA Ca-11, Kyoto, Japan). The initial Ca$^{2+}$ concentration of the solution is 460 ppm.

(2) All samples were dried and weighed prior to testing. Then, the volume and permeability of the cylinder specimens were measured,

(3) All samples were placed into the hydrothermal reactor, and 800 mL seawater was added. All samples were immersed by the solution. Then, CO$_2$ was injected into the reactor by the syringe pump. The pressure and temperature were held at 7MPa and 30 °C, respectively.

(4) The solution was analyzed for dissolved Ca$^{2+}$ concentration during the reaction. After 3 days of reaction time, the specimens were taken out and washed repeatedly with distilled water to avoid influence from salt crystals after drying. Then, they were oven-dried to study the changes of mass. The effective volume and permeability of the cylinder specimens were tested by the methods introduced above. One of the cube specimens was taken and prepared for SEM analysis.

(5) The specimens (except the sample analyzed by SEM) were placed back into the reactor, and the same amount of seawater was added to continue the next stage of reaction. Then, steps (3)–(4) were repeated.

(6) After 15 days, all the samples were taken out from the reactor to investigate the evolution of the dry mass, effective volume, permeability, and the change of surface properties and mechanical behavior.

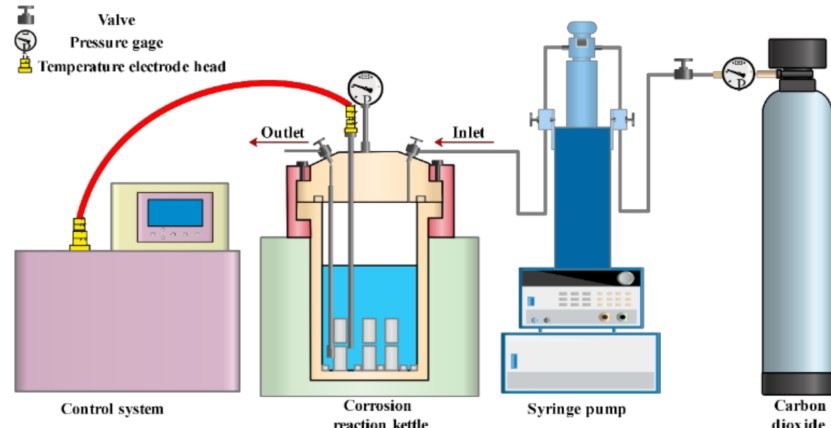

**Figure 5.** Schematic diagram of the hydrothermal reactor system: pressure up to 30 MPa and temperature up to 100 °C. The inlet in the reaction kettle allowed CO$_2$ in. The outlet with a filter enabled liquid sampling.

For the specimens of Group control, the same amount of seawater was used, and the temperature was kept at the constant temperature of 30 °C by a thermostatic water bath. To account for time-dependent effects, the cube samples of Group control were taken out to investigate the evolution of the dry mass every 3 days. After 15 days, all samples were taken out to investigate the changes of the dry mass, and the effective volume and permeability of the cylinder samples were tested. The last cube sample was observed by SEM. In addition, mechanical tests were conducted.

2.2.5. Mechanical Tests

The uniaxial compression and split tensile tests were performed on the RMT multifunctional rock rigidity testing machine, which was independently developed by the State Key Laboratory of Geomechanics and Engineering, Wuhan Institute of Rock and Soil Mechanics, Chinese Academy of Sciences, as shown in Figure 6. The tests were conducted according to the DL/T5368-2007 Standards of Rock Experiment procedures of Water Conservancy and Hydropower Engineering of the Chinese Ministry of Water Resources (2007) [31]. The samples were saturated before the test, and the loading protocol was displacement-controlled with a displacement rate of 0.001 mm/s.

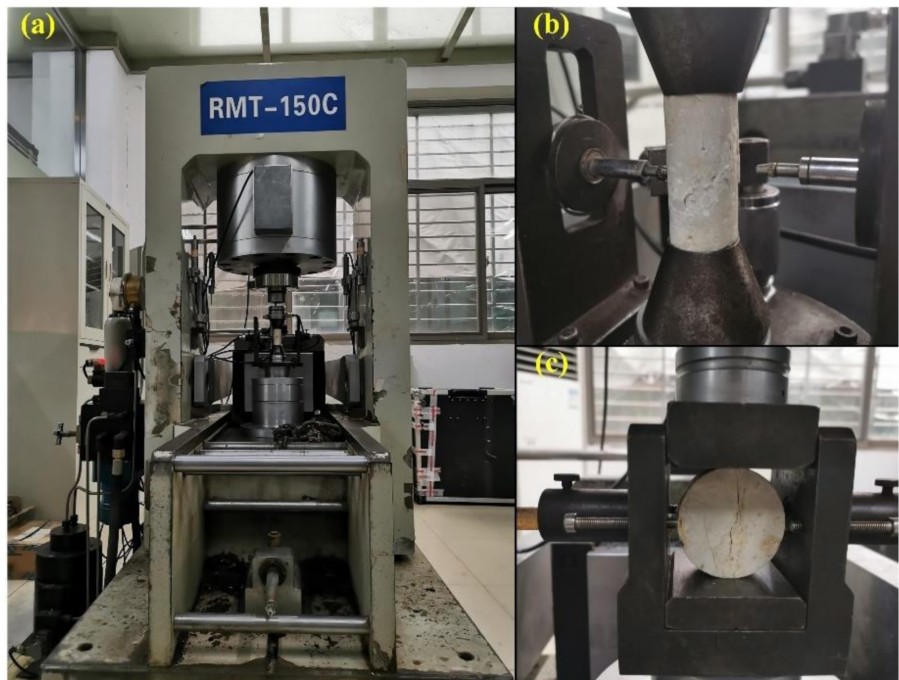

**Figure 6.** (**a**) The RMT multifunctional rock rigidity testing machine, (**b**) the uniaxial compression test, (**c**) the split tensile test.

## 3. Results and Discussion

### 3.1. Changes of Physical Properties

3.1.1. Mass Loss

The final changes of the dry mass and water absorption coefficient of the specimens from the two groups after 15 days of reaction time are summarized in Table 1. Figure 7 shows the decrease in the percentage of the mass of the rock samples with the increasing reaction time, and the error bars in the figure present the standard deviations. The mass loss of the samples of Group $CO_2$ was much more significant than those of Group control throughout all reactions due to the existence of $CO_2$. By the end of the reaction, the mass loss of the samples of Group $CO_2$ was negligible, while the average mass loss of the specimens (except for the BS) in Group control was about 1.03 g. Figure 8 shows the change of concentration of $Ca^{2+}$ of the solution of two groups with reaction time. The concentration

of $Ca^{2+}$ of Group control was unchanged during testing, while for Group $CO_2$, it increased gradually within 6 h and remained constant afterward. The evident significance of the concentration of $Ca^{2+}$ indicates that the chemical dissolution played an important role in mass transport. As the reef limestone is mainly composed of calcite, the main reaction, which caused the dissolution, is as follows [35]:

$$CO_2 + H_2O + CaCO_3 \rightleftharpoons Ca(HCO_3)_2 \tag{4}$$

The calcium bicarbonate is quite unstable, and the chemical reaction in the reactor may be a dynamic equilibrium.

**Table 1.** The final mass changes of the specimens from the two groups after the reaction.

| Group | Test Condition | Number of Sample | Form | Water Absorption Coefficient (%) | Mass (g) | | |
|---|---|---|---|---|---|---|---|
| | | | | | Before | After | ΔM |
| CO₂ | Injection of CO₂ Pressure of 7 Mpa 30 °C | 1-1 | BUC | 0.71 | 64.04 | 63.36 | −0.68 |
| | | 1-2 | | 0.88 | 63.59 | 62.86 | −0.73 |
| | | 1-3 | | 0.78 | 65.57 | 64.82 | −0.75 |
| | | CT1-2 | | 0.71 | 128.46 | 126.85 | −1.60 |
| | | B1-1 | BUT | 0.86 | 126.65 | 125.31 | −1.34 |
| | | B1-2 | | 0.61 | 127.71 | 126.62 | −1.09 |
| | | V | BS | 1.10 | 2.74 | 2.64 | −0.10 |
| Control | Without CO₂ Constant temperature 30 °C | 1-4 | buc | 1.14 | 60.52 | 60.50 | −0.02 |
| | | 1-5 | | 1.06 | 60.30 | 60.24 | −0.06 |
| | | 1-6 | | 1.17 | 59.72 | 59.70 | −0.02 |
| | | b1-1 | but | 0.67 | 126.83 | 126.78 | −0.05 |
| | | b1-2 | | 0.56 | 125.74 | 125.60 | −0.14 |
| | | b1-3 | | 0.56 | 125.21 | 125.09 | −0.12 |
| | | 11 | bs | 1.38 | 2.52 | 2.52 | 0.00 |

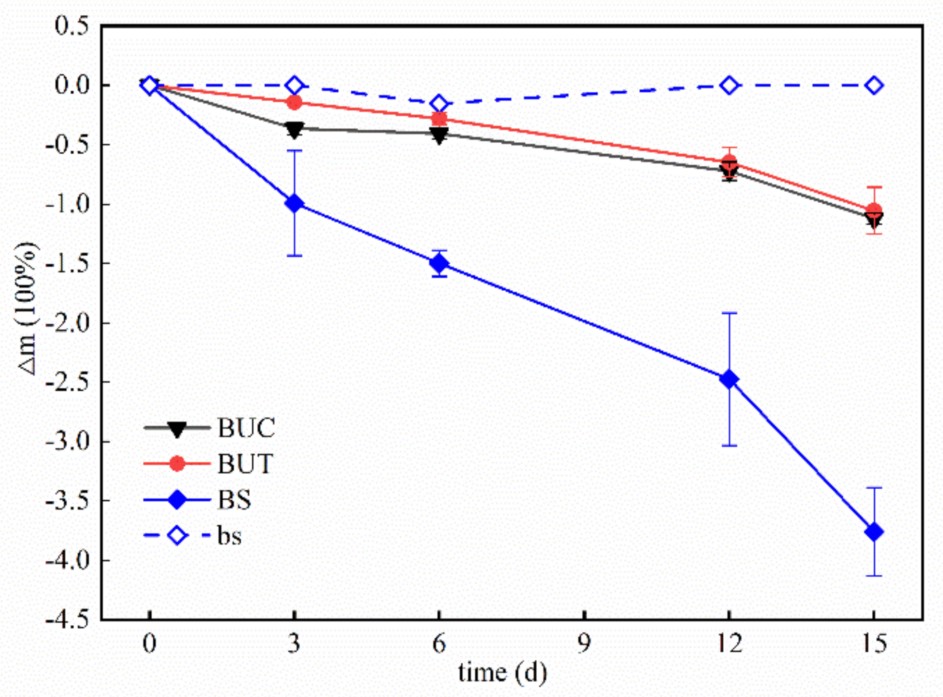

**Figure 7.** The changes of the mass of the specimens from the two groups with increasing reaction time.

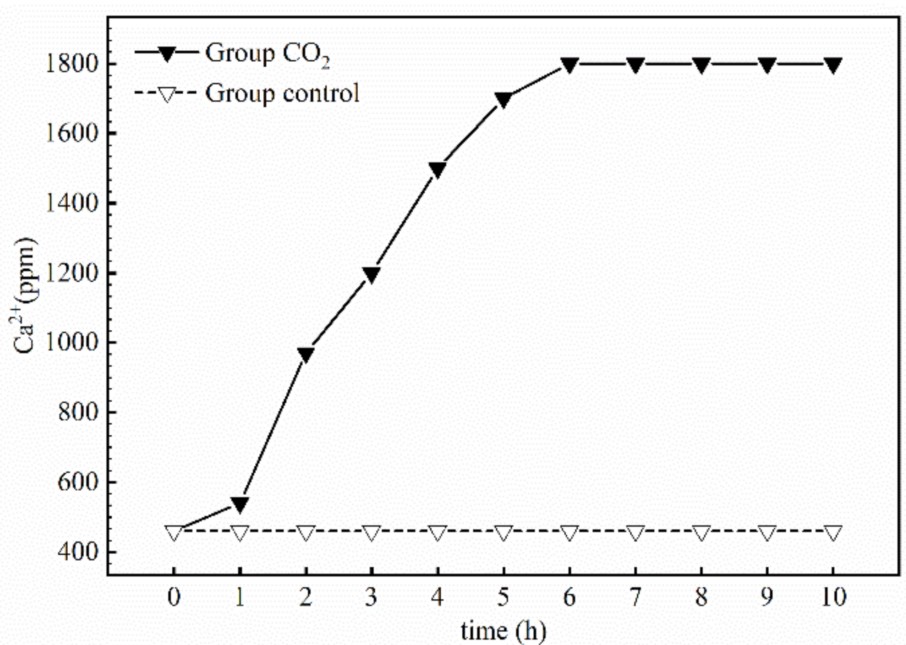

**Figure 8.** Change of $Ca^{2+}$ concentration of the solutions from the two groups with time.

Further analysis of the dates in Figure 7 reveals that the percentages of mass loss for the specimens of Group $CO_2$ were around 1.05%, 1.12% and 3.76% for the BUT, BUC and BS samples, respectively. This means that the smaller the sample, the higher the percentage of mass loss. The percentage of mass loss is believed to be influenced by the specific surface area of the sample. The smaller particles have a higher specific surface area, which will become the larger than the contact area with the reaction solution [36]. Therefore, the smaller the sample, the higher the degree of the chemical reaction of $CO_2$–seawater–rock. Referring to the dates of the water absorption coefficient ($W_a$) in Table 1, it can be seen that the order of the $W_a$ of the samples from large to small was BS (1.10), BUC (0.79) and BUT (0.73), respectively. The larger $W_a$ also indicates more solution consumption, which further verifies the results in Figure 7. Figure 7 also presents that the mass loss of all types of samples in Group $CO_2$ continued to increase after each reaction period, and the dissolution rate increased with the time. This was because that the reactive surface area followed a growing trend with reaction time [37]. This is discussed in further detail below.

3.1.2. Changes in Volume, Porosity and Permeability

Table 2 shows the final changes of the effective volume and porosity of the cylinder samples from the two groups after the reaction. The permeability of the samples from the two groups at different reaction times is summarized in Table 3. $P_c$ is the confining pressure when the permeability was measured. It is clear that the volume, porosity and permeability of the specimens of Group control barely changed during the test, which is comparable to the results of mass. After 15 days, the volume of the specimens of Group $CO_2$ exhibited a minor reduction (about 0.14–0.79 $cm^3$) in volume, which caused a 0.57–3.15% increase in the porosity. However, the permeability of the samples of Group $CO_2$ showed a significant increase regardless of the limited volume change of the samples. The permeability also increased exponentially in time. For example, the initial permeability for the 1-1 sample was 0.65 mD when the $P_c$ was 1 MPa, but it increased to 11.40 mD at the end of the trial. Hence, although a tiny amount of rock dissolved during the reaction, it had a tremendous impact on the permeability.

**Table 2.** The final changes in the effective volume and porosity of the cylinder samples from the two groups after 15 days of reaction time.

| Group | Number of Sample | Size | Volume(cm$^3$) | | | Porosity (%) | | |
|---|---|---|---|---|---|---|---|---|
| | | | Before | After | ΔV(cm$^3$) | Before | After | ΔP |
| CO$_2$ | 1-1 | BUC | 24.45 | 23.66 | −0.79 | 3.05 | 6.20 | 3.15 |
| | 1-2 | | 23.59 | 23.45 | −0.14 | 6.98 | 7.54 | 0.57 |
| | 1-3 | | 24.53 | 24.19 | −0.34 | 2.96 | 4.31 | 1.34 |
| Control | 1-4 | buc | 22.82 | 22.76 | −0.02 | 5.32 | 5.58 | 0.26 |
| | 1-5 | | 22.56 | 22.52 | −0.06 | 5.63 | 5.82 | 0.19 |
| | 1-6 | | 22.29 | 22.24 | −0.02 | 7.70 | 7.90 | 0.21 |

**Table 3.** The permeability of the samples from the two groups at different reaction days.

| Group | Number of Sample | $P_c$ (MPa) | Permaebility (mD) | | | | |
|---|---|---|---|---|---|---|---|
| | | | 0 Day | 3 Day | 6 Day | 12 Day | 15 Day |
| CO$_2$ | 1-1 | 1 | 0.65 | 2.51 | 2.29 | 5.92 | 11.40 |
| | | 3 | 0.48 | 1.68 | 1.55 | 3.65 | 8.83 |
| | | 5 | 0.28 | 0.91 | 0.86 | 1.94 | 5.56 |
| | | 7 | 0.12 | 0.34 | 0.38 | 0.94 | 3.29 |
| | 1-2 | 1 | 3.28 | 13.52 | 9.12 | 26.39 | 24.47 |
| | | 3 | 2.48 | 8.57 | 6.31 | 15.22 | 16.36 |
| | | 5 | 1.61 | 4.80 | 3.44 | 7.73 | 9.86 |
| | | 7 | 0.84 | 2.83 | 1.60 | 3.82 | 5.31 |
| | 1-3 | 1 | 0.06 | 1.47 | 0.82 | 2.68 | 5.83 |
| | | 3 | 0.07 | 0.87 | 0.43 | 1.50 | 2.47 |
| | | 5 | 0.06 | 0.48 | 0.21 | 0.77 | 2.07 |
| | | 7 | 0.02 | 0.24 | 0.10 | 0.43 | 1.18 |
| Control | 1-4 | 1 | 0.45 | - | - | - | 0.45 |
| | | 3 | 0.21 | - | - | - | 0.21 |
| | | 5 | 0.09 | - | - | - | 0.09 |
| | | 7 | 0.05 | - | - | - | 0.05 |
| | 1-5 | 1 | 0.36 | - | - | - | 0.35 |
| | | 3 | 0.14 | - | - | - | 0.14 |
| | | 5 | 0.08 | - | - | - | 0.07 |
| | | 7 | 0.04 | - | - | - | 0.03 |
| | 1-6 | 1 | 0.27 | - | - | - | 0.26 |
| | | 3 | 0.11 | - | - | - | 0.11 |
| | | 5 | 0.06 | - | - | - | 0.05 |
| | | 7 | 0.03 | - | - | - | 0.03 |

To obtain further insights into the process of the CO$_2$–seawater–limestone reaction, we carried out a detailed analysis of the results in Group CO$_2$. Figure 9 shows the changes of the effective volume, porosity and permeability of the BUT samples with increasing reaction time. The changes of the volume and porosity were presented as percentages, while the changes of permeability were presented as differences. The error bars in the figure present the standard deviations. In Figure 9, the trend of the permeability with time was similar to the trends of the porosity with time under all confining pressures. The relationship may be explained by the fact that permeability is an intrinsic property of the rock that depends mainly on the effective porosity, and it may experience a bigger change corresponding to a slight alteration of the porosity [38–41]. Further, the smaller the confining pressure, the easier the seepage, and the higher the permeability [42].

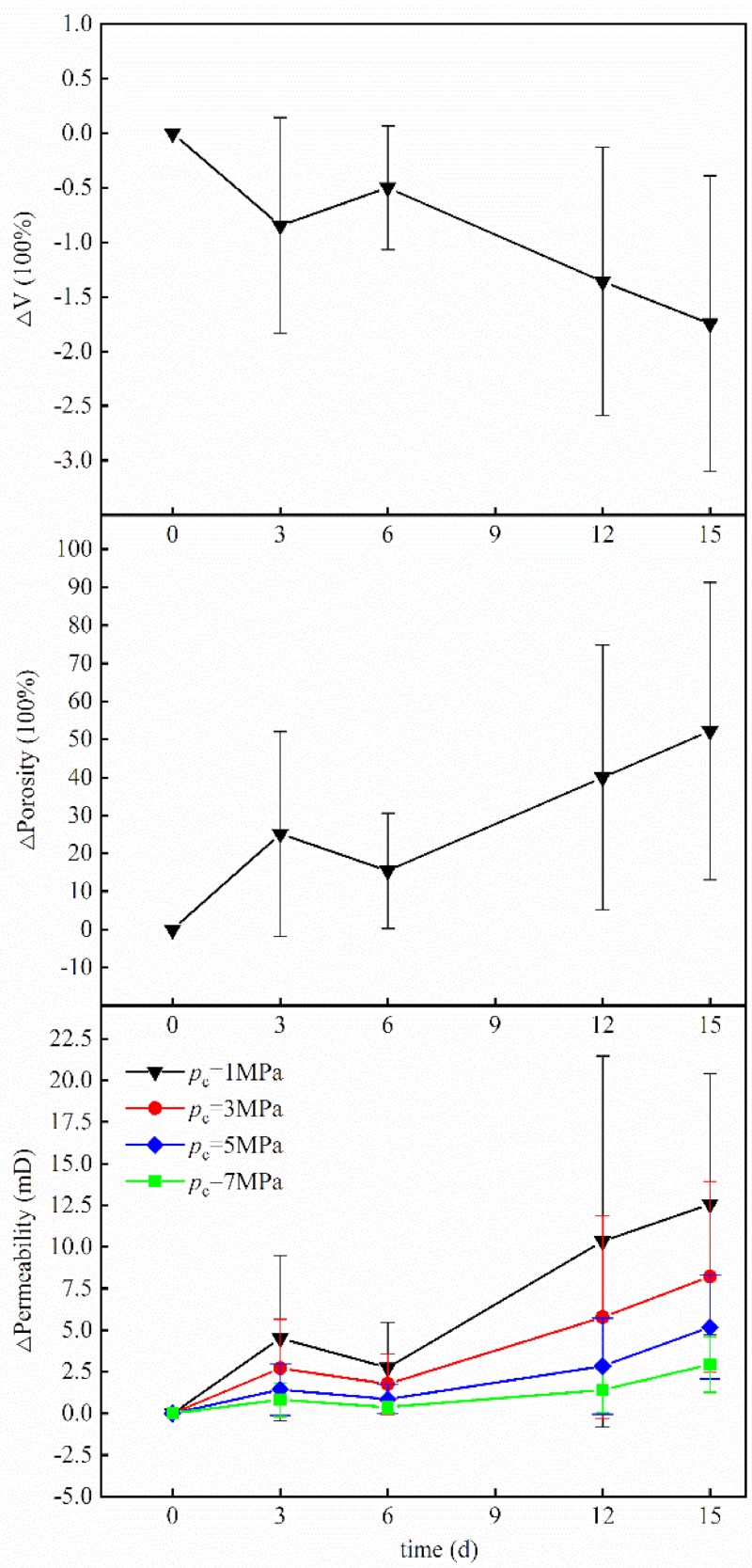

**Figure 9.** The changes of the volume, porosity and permeability of the BUC samples with increasing reaction time.

The volume showed a general decreasing trend with time because of the mineral dissolution, while the porosity showed an increasing trend, which caused the increase of permeability. However, the result of the reaction after 3–6 days of reaction time was different. After 6 days of reaction time, the volume of the samples increased slightly, which may have been induced by the precipitation of $CaCO_3$ (shown in Figure 10a). Figure 10 shows the evolution of the surface texture of the cube samples from the two groups (bs and BS) by SEM. Figure 10a shows the greyscale images of the SEM of Group $CO_2$, and Figure 10b shows the binary images. The image of Group control after the reaction at 15 d is shown in Figure 10c. After 15 days of soaking in seawater without extra $CO_2$, the surface of the bs sample was still as smooth as before the test. However, the surface of BS became rough after only 3 days because of the dissolution of the calcite under the $CO_2$–seawater–limestone reaction. In the subsequent analysis, the quantification of basic size parameters (such as pore count, pore area, Feret diameter and surface porosity of the corrosion pores) of the corrosion pores in Figure 10b were analyzed using the built-in measurement tools of the ImageJ software. The results are shown in Figure 11 and Table 4. These results indicate that the size of surface corrosion pores increased as the reaction time increased.

From the results above, the mechanism of chemical dissolution that induced microstructure evolution and changes of the physical property can be revealed. In the early stage of the reaction (during days 0–3), the reef limestone dissolved into calcium bicarbonate, and a large number of micropores formed. The average size of micropores was about 0.15 $\mu m^2$, and the corrosion porosity of the surface was about 6.08% after 3 days of reaction time. As the reaction continued, the samples continued to dissolve, which led to the enlargement of the micropores, some of which merged together and formed bigger wormholes (shown in Figure 10b, the date of Group $CO_2$ is 6 day). The average size of wormholes was about 0.74 $\mu m^2$, and the corrosion porosity of the surface increased to 17.73%. The sample surface observed under SEM showed dissolution-precipitation features during the reaction (seen in Figure 10a, the date of Group $CO_2$ is 6 day) because the calcium bicarbonate was quite unstable. The secondary minerals calcite usually settles on the surface of the sample and may clog some micropores, which results in a tiny decrease of the porosity, enhancing its ability to serve as a fluid barrier, and the permeabilities slightly decrease after 6 days [43,44] (in Figure 9). However, the influences of the precipitates are likely to be transient. According to the results in all figures and tables, the $CO_2$–seawater–rock chemical reaction processes were dominated by the dissolution. There was only a small number of visible precipitates on the surface on day 6. The formed crystals were not very stable and would have been dissolved by an influx of new water [45]. Then pores became larger, connected with each other, and further developed into wider pore channels or fractures (seen in Figure 10, dates of Group $CO_2$ are 12 day and 15 day). Therefore, the permeability of the rock samples improved again because of the new percolation channels.

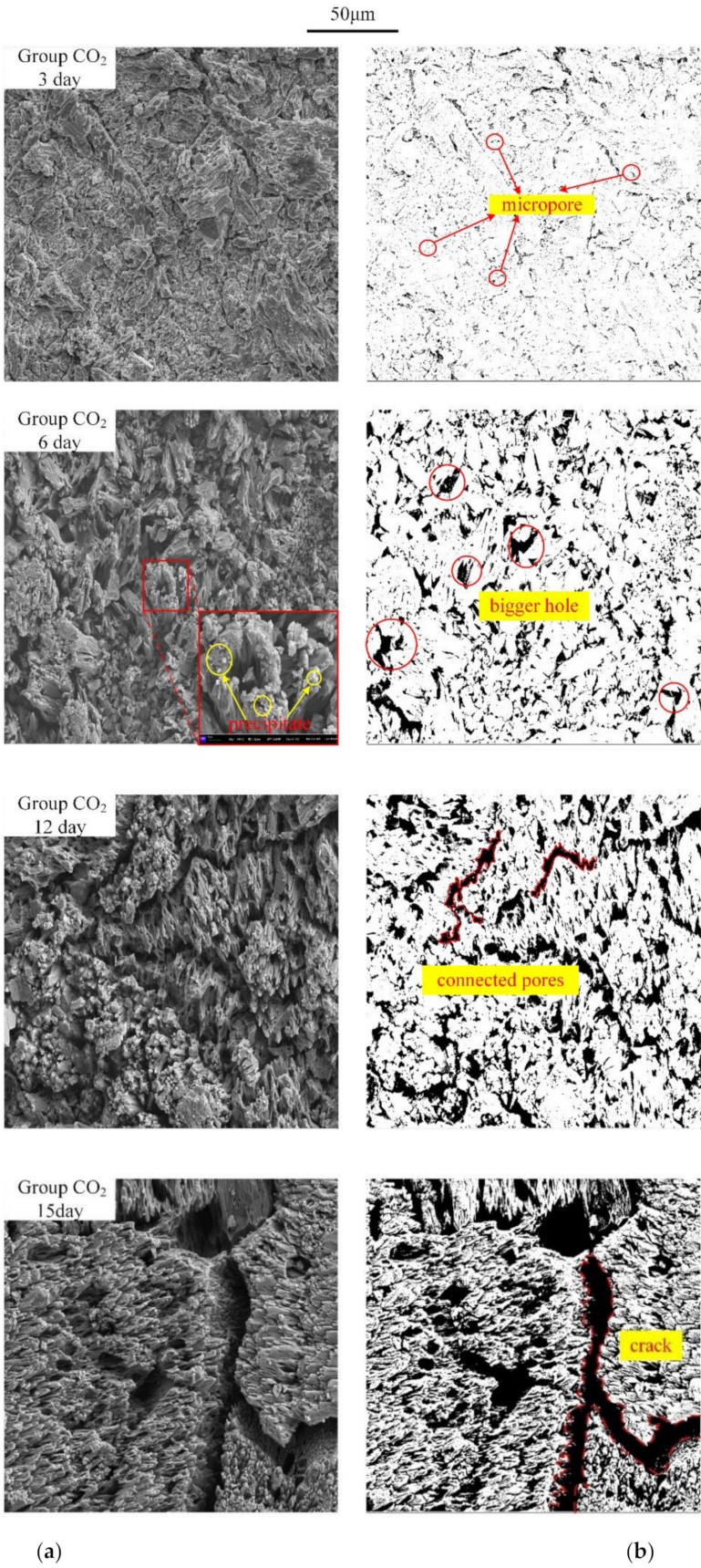

(**a**)  (**b**)

**Figure 10.** *Cont.*

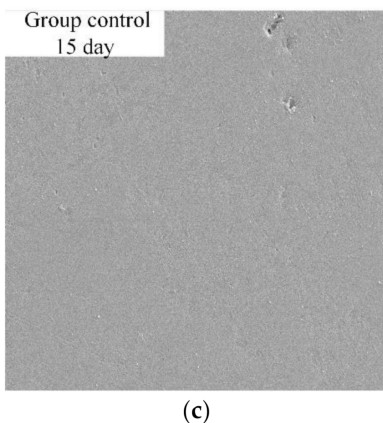

(**c**)

**Figure 10.** Scanning electron microscopy images of the cube samples: (**a**) Corrosion evolution of the Group $CO_2$ for 15 days, showing dissolution after the reaction at 3 day, 6 day, 12 day and 15 day, respectively. There are few $CaCO_3$ precipitates in the image of 6 day. (**b**) The corresponding binary images of (**a**), with the rock matrix indicated in white and corrosion pores in black. (**c**) The image of Group control after the reaction at 15 day, showing no dissolution. The scale bar applies to all images.

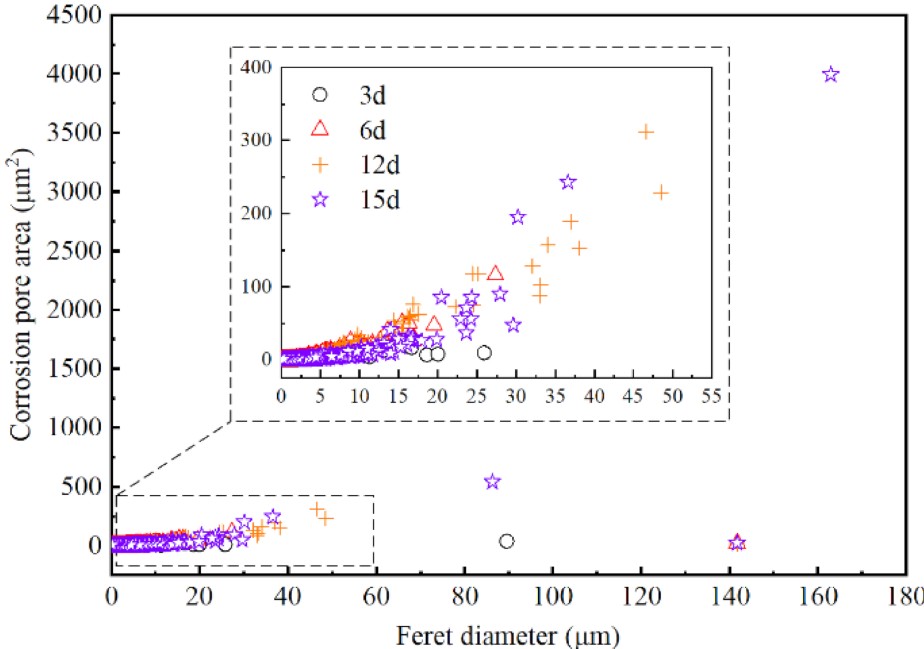

**Figure 11.** Corrosion pore size distribution of surface of the BS samples in Group $CO_2$. Black circles are the pores of sample at 3 day, red triangles are the pores of sample at 6 day, green crosses are the pores of sample at 12 day and purple pentagrams are the pores of sample at 15 day, respectively.

**Table 4.** Probability statistics table of the corrosion pores of the BS samples.

| Reaction Time (Day) | Pore Count | Average Size ($\mu m^2$) | Corrosion Area ($\mu m^2$) | Surface Porosity (%) |
|---|---|---|---|---|
| 3 | 11,897 | 0.15 | 1837.92 | 6.08 |
| 6 | 3388 | 0.74 | 2491.90 | 17.73 |
| 12 | 3381 | 1.20 | 4043.15 | 28.76 |
| 15 | 5059 | 1.38 | 6979.61 | 49.65 |

### 3.2. Change of Mechanical Property

In order to study the influence of chemical interaction on the mechanical properties of the coral reef limestones, uniaxial compression and Brazilian splitting experiments were conducted. Tables 5 and 6 summarize the results of the uniaxial compression and tensile tests of the specimens, respectively. The average compressive strength and average elastic modulus of Group $CO_2$ was 44.65 MPa and 4.56 GPa, respectively. The average compressive strength and average elastic modulus of Group control was 52.14 MPa and 4.6 GPa, respectively. The results seem to suggest that the average peak strength and elastic modulus of the samples reacted with the $CO_2$–seawater reaction were slightly lower than the samples only soaked by seawater without extra $CO_2$. However, the results of the tensile tests are opposite to that of Table 6. The average tensile strength of Group $CO_2$ (7.04 MPa) was slightly higher than that of Group control (6.42 MPa).

**Table 5.** Results of the uniaxial compressive strength test of the saturated specimens from two groups after 15 days of reaction time.

| Group | Number of Sample | Form | Compressive Strength (MPa) | Average Compressive Strength (MPa) | Elastic Modulus (GPa) | Average Elastic Modulus (GPa) |
|---|---|---|---|---|---|---|
| $CO_2$ | 1-1 | BUC | 48.57 | 44.65 | 4.53 | 4.56 |
| | 1-2 | | 23.52 | | 3.54 | |
| | 1-3 | | 61.87 | | 5.62 | |
| Control | 1-4 | buc | 63.38 | 52.14 | 4.59 | 4.60 |
| | 1-5 | | 49.35 | | 6.04 | |
| | 1-6 | | 43.68 | | 3.16 | |

**Table 6.** Results of the uniaxial tensile strength test of the saturated specimens from two groups after 15 days of reaction time.

| Group | Number of Sample | Form | Tensile Strength (MPa) | Average Tensile Strength (MPa) |
|---|---|---|---|---|
| $CO_2$ | CT1-2 | BUT | 6.94 | 7.04 |
| | B1-1 | | 6.50 | |
| | B1-2 | | 7.67 | |
| Control | b1-1 | but | 5.86 | 6.42 |
| | b1-2 | | 5.75 | |
| | b1-3 | | 7.66 | |

Figure 12 shows the stress–strain curve of the saturated samples of two groups under uniaxial compressive condition after 15 days of reaction. The uniaxial compression strength of the samples in this study was within the range from 23 to 65 MPa. There was no significant difference between the samples in the two groups on the mechanical response. All the samples presented the features of brittle failure. As shown in Figure 13, both groups of samples showed tensile failure, and the failure cracks spread along the large sample pores. Several studies have shown that coral reef limestones always show inhomogeneous characteristics. Although all samples were taken side-by-side from the same core, the mechanical properties of them may be different [29,46]. Moreover, their uniaxial compression strength may be quite small compared to that of the other rocks, and the values can vary significantly [47]. Therefore, if the strength change is small, it is difficult to see a significant difference in the mechanical results.

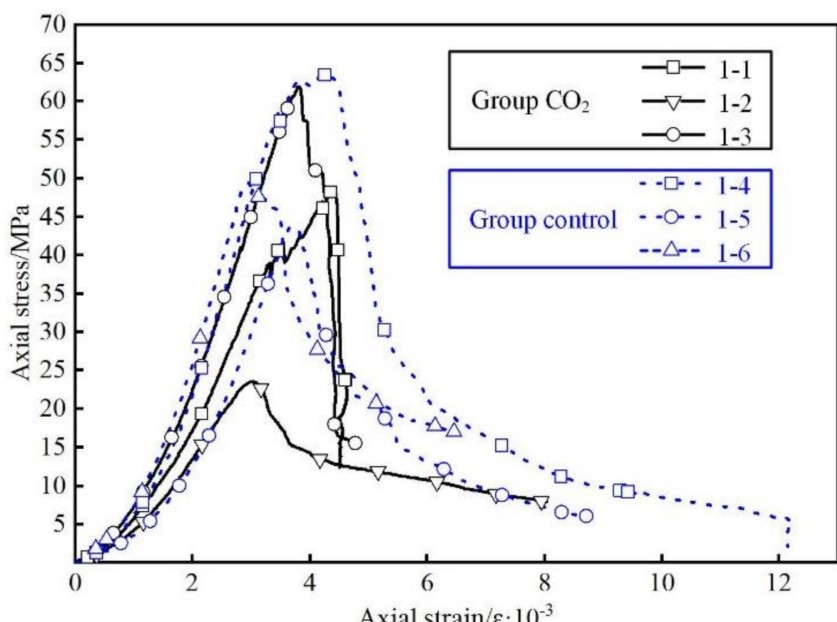

**Figure 12.** The stress–strain curve of the saturated sample under the uniaxial compressive condition of the two groups after 15 days of reaction time.

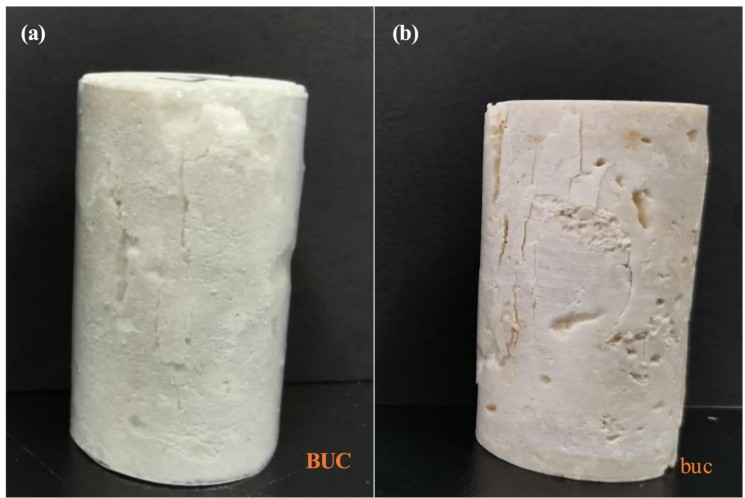

**Figure 13.** Samples of two groups after the uniaxial compression test: (**a**) 1-1 sample of Group $CO_2$, (**b**) 1-4 sample of Group control.

In order to further explore the effect of seawater acidification on the mechanical properties of reef limestone, the CT scanning results of the uniaxial compression sample 11 were analyzed. Figure 14 shows the 3D reconstructed images of the 1-1 sample before and after the reaction, and it can be found that the surface became rough and small corrosion pits appeared after reaction. Figure 15 shows the porosity changes of the 1-1 sample before and after the reaction based on the CT scan. The porosity of the sample was analyzed by the Avzio Software 2020.2 (Thermo Fisher Scientific, MA, USA). It can be found that the original porosity of the sample was 1.48%. After the reaction, the sample dissolved, and the porosity increased to 2.1% (shown in Figure 15c). However, the results of the CT analysis depend on the scanning resolution of the samples. That might be the reason why the porosity of the sample is smaller than that in Table 2 (the original porosity was 3.01%, and the porosity after the reaction was 6.20%). An increasing trend of the sample porosity with reaction time is consistent.

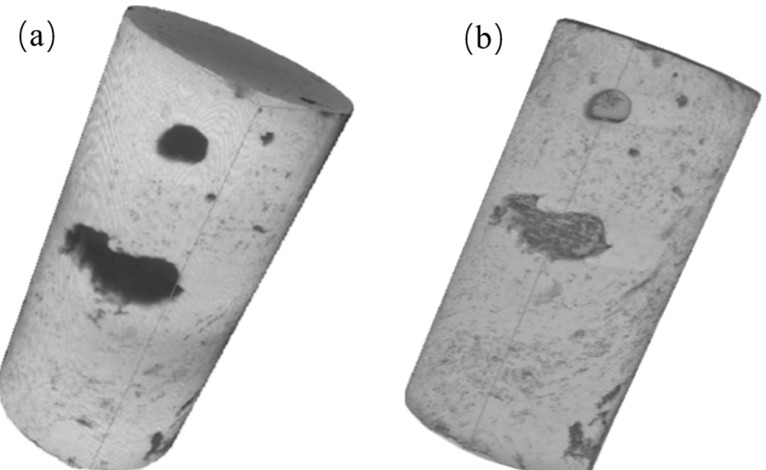

**Figure 14.** The 3D reconstructed images of the 1-1 sample before and after reaction: (**a**) the image of the sample before the reaction, (**b**) the image of the sample after the reaction.

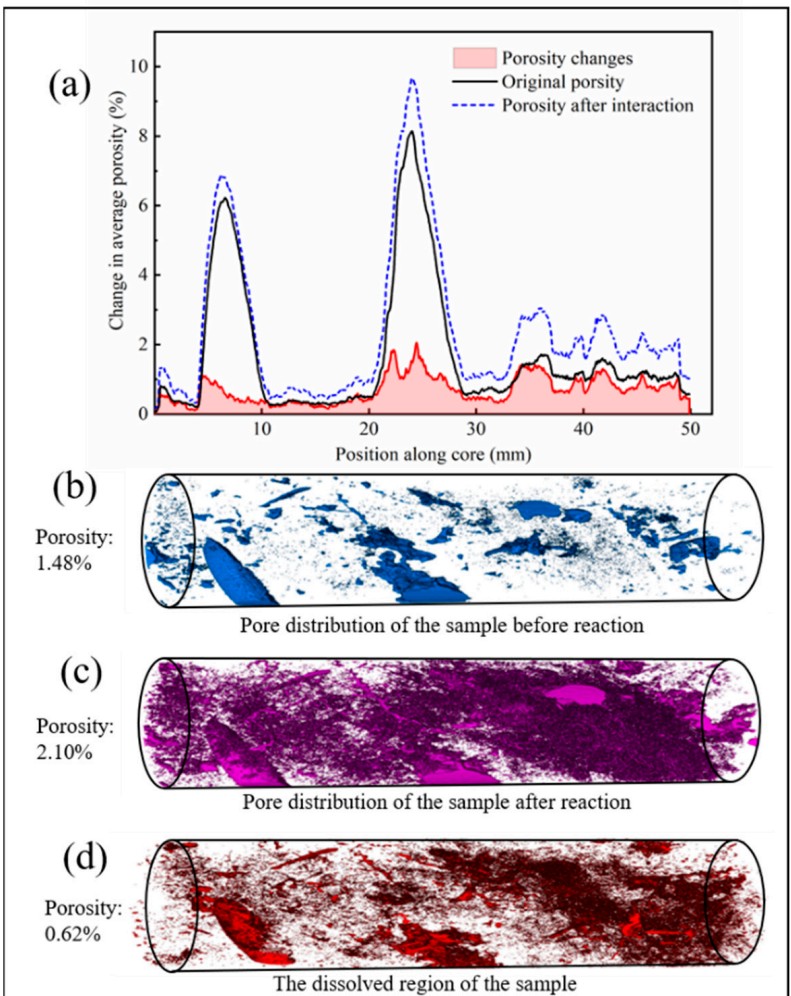

**Figure 15.** The porosity changes of the 1-1 samples before and after the reaction according to the CT scan results. (**a**) The porosity changes of CT slices lay by lay. (**b**) The pore distribution of the sample before the reaction. The average porosity analyzed by the CT results was 1.48%. (**c**) The pore distribution of the sample after the reaction. The average porosity analyzed by the CT results was 2.10%. (**d**) The dissolved region of the sample. The porosity analyzed by the CT results was 0.62%.

The results are similar to those of other studies showing that although the carbon dioxide environment changes the microstructure of rocks, it has little effect on their mechanical properties [48]. This is because the total amount of dissolved in the whole reaction process is very small, and the sample structure does not significantly change. However, the reaction rate is very fast, and the calcium ions in the solution reach a dynamic equilibrium in a few hours (shown in Figure 8). As the reaction occurs in a limited container, the dissolution is difficult to continue because the solution that could not be timely supplemented. The results of the SEM revealed that the micropores in the sample expanded rapidly as the reaction proceeded. It can be seen from Figure 15 that the corrosion area of the sample expanded along the original pores of the sample and the pores grew bigger after the reaction. Liu et al. [49] found that the mechanical properties of reef limestone are closely related to porosity. Figure 16 shows the 3D visualizations of the 1-1 sample after the uniaxial compressive test, revealing that tensile fractures occurred along the pores of the sample when the sample was broken. When the dissolved pores are large enough, the mechanical properties of reef limestones may be influenced by the reaction with acidified seawater. Therefore, the effect of acidified seawater on mechanics remains unclear and requires further study and discussion.

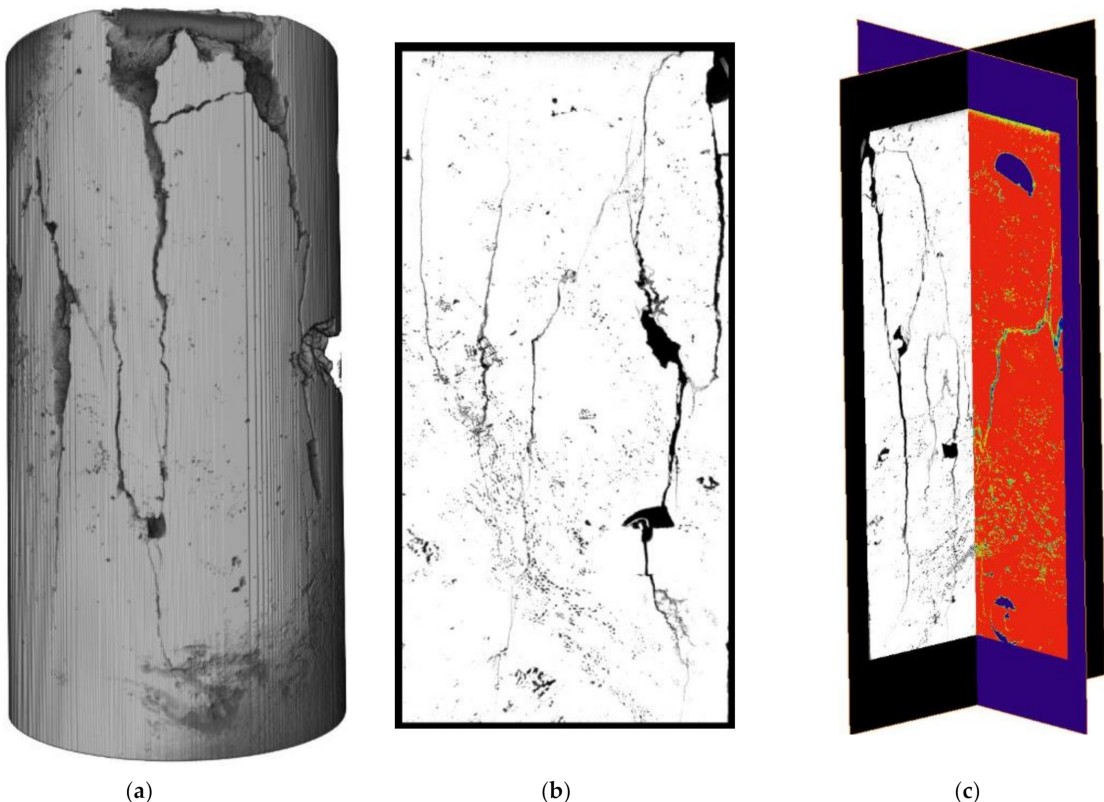

(**a**)　　　　　　　　　　(**b**)　　　　　　　　　　(**c**)

**Figure 16.** Three-dimensional visualizations of the 1-1 sample after the uniaxial compressive test, (**a**) raw image, (**b**) image of the longitudinal sections of the sample, (**c**) image of the longitudinal sections of the sample in two directions.

## 4. Conclusions

The effects of $CO_2$–seawater on the alteration of physical and mechanical properties of coral reef limestone were analyzed through an experiment conducted in a hydrothermal reactor. Based on the data and information generated, the following conclusions were drawn:

(1)　The coral reef limestones dissolved when immersed in seawater supersaturated with carbon dioxide, which caused the changes of the physical properties of the rock

samples. The mass and volume of reef limestones decreased during the reaction, and the permeability increased during the reaction process.

(2) Although the reaction was mainly a dissolution process, there was a small amount of precipitation produced after 15 days of reaction time, which had a big impact on the permeability of the rock.

(3) As the reaction time increased, the reaction rate increased, and the microscopic pore structure of the reaction surface changed significantly during the reaction process. At the beginning of the reaction, a large amount of micropores were generated. Then, the micropores grew bigger and connected with each other as the reaction continued. In this way, the reaction contact surface increased, making the reaction rate increase with time.

(4) Because the rocks were soaked for only 15 days, the total cumulative amount of calcium carbonate dissolved was less, and the mechanical properties were not affected.

**Author Contributions:** Conceptualization, Y.Z. and Q.L.; methodology, Y.Z.; validation, Q.L. and R.W.; formal analysis, Y.Z.; investigation, Y.Z.; resources, R.W.; data curation, Q.L. and T.Y.; writing—original draft preparation, Y.Z.; writing—review and editing, Q.L. and T.Y.; visualization, Y.Z.; supervision, R.W.; project administration, R.W.; funding acquisition, R.W. All authors have read and agreed to the published version of the manuscript.

**Funding:** This research was funded by the National Natural Science Foundation of China, grant number 41330642 and 41772337.

**Institutional Review Board Statement:** Not applicable.

**Informed Consent Statement:** Not applicable.

**Data Availability Statement:** Not applicable.

**Conflicts of Interest:** The authors declare no conflict of interest.

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
