# Peer review of "Changes of Physical and Mechanical Properties of Coral Reef Limestone under CO2–Seawater–Rock Interaction"

_applsci, doi:10.3390/app12094105_

Round 1
Reviewer 1 Report
Ref. applsci-1680900-peer-review-v1
Changes of physical and mechanical properties of coral reef limestone under CO2-seawater-rock interaction
by
Yu Zhong, Qi Li, Ren Wang, Ting Yao
The manuscript deals with the physical and chemical interaction between CO2, seawater and framework structure reef limestone. The paper is well organised, and the results are sound. It could be accepted in this form, but It seems to me that several typos should be fixed. In the attached file there are also some suggestions for improving the text.

Reviewer 2 Report
Figure 3
Please compare the surface in the same magnification ratio or write clearly in the figure or body text that the magnification ratio is different from each other.
Figures 4 and 5
I do not think them necessary.
Table 1, 2 , 4, 5, 6
Please modify the number of significant figures considering your experimental results.
Group name
I do not think naming groups 1 and 2 are simple to understand.
There are only with/without injection CO2.
For example, "CO2" and "control" or "with CO2" and "without CO2",,,
There should be much better naming.
Figure 11
Where is the scale ber?
Figure 12
I do not think optic green is easily viewable.
Figure 13
I do not think these colour legends are simple to understand.
Figure 15
What is the vertical value for the original porosity and the porosity after the interaction?
It would be the porosity in the section, but not the change in average porosity.
Table 2, Figure 10, Table 4, Figure 15
Are the porosity values comparable to each other?
Please describe a discussion about that.
